# Research on IMU-Assisted UWB-Based Positioning Algorithm in Underground Coal Mines

**DOI:** 10.3390/mi14071481

**Published:** 2023-07-24

**Authors:** Lei Wang, Shangqi Zhang, Junyan Qi, Hongren Chen, Ruifu Yuan

**Affiliations:** 1School of Computer Science and Technology, Henan Polytechnic University, Jiaozuo 454000, China; wlqjy@hpu.edu.cn; 2School of Software, Henan Polytechnic University, Jiaozuo 454000, China; 3School of Surveying and Land Information Engineering, Henan Polytechnic University, Jiaozuo 454000, China; chr18737027727@163.com; 4School of Energy Science and Engineering, Henan Polytechnic University, Jiaozuo 454000, China; yrf@hpu.edu.cn

**Keywords:** UWB positioning system, shearer, MEMS IMU, ESKF

## Abstract

The application of an ultra-wideband (UWB) positioning system in a Global Positioning System (GPS) denial environment such as an underground coal mine, mainly focuses on position information and rarely involves information such as direction attitude. Position accuracy is often affected by multipath, non-visible ranges, base station layout, and more. We proposed an IMU-assisted UWB-based positioning system for the provision of positioning and orientation services to coal miners in underground mines. The Error-State Kalman Filter (ESKF) is used to filter the errors in the measured data from the IMU-assisted UWB positioning system to obtain the best estimate of the error for the current situation and correct for inaccuracies due to approximations. The base station layout of the IMU-assisted UWB positioning system was also simulated. The reasonable setting of the reference base station location can suppress multi-access interference and improve positioning accuracy to a certain extent. Numerous simulation experiments have been conducted in GPS denial environments, such as underground coal mines. The experimental results show the effectiveness of the method for determining the position, direction, and attitude of the coal miner under the mine, which provides a better reference value for positioning and orientation in a GPS rejection environment such as under the mine.

## 1. Introduction

Coal is the main energy source in China [1]. The application of intelligent unmanned mining [2,3] technology to improve the level of coal mine equipment is of great importance to realize less humanized operation, unmanned mining, and reduce labor costs [4]. One of the most critical is the implementation of the linkage of the coal mining machine, scraper conveyor, and hydraulic support [5] for the comprehensive mining face. Memory cutting, hydraulic brace movements, and mining process choices are all based on the coal mining machine’s precise location. The micro-electromechanical inertial measurement unit (MIMU) [6,7] provides a wealth of navigation information for the Strapdown Inertial Navigation System (SINS) [8] with its low cost and small size. However, the MIMU orientation method is less accurate because it is more subject to ambient interference and mistakes brought on by attitude and position solving from pure inertial guiding data. Where conventional GPS [9] cannot be used in underground mine environments [10], UWB [11,12] positioning systems can be used in environments where GPS is denied. SINS has been successfully applied to the positioning of continuous coal mining machines [13,14,15].

The reference [16] sets motion limits on the miner to increase positioning precision, while the reference [17,18] optimally estimates the closure path of a coal miner engaged in longwall mining. SINS/LiDAR [19], SINS/UWB [20], SINS/WSN [21], and other typical combined positioning systems are used to address the position error accumulation issue with SINS. The development of UWB [22] positioning systems in indoor [23,24,25], unmanned aerial vehicles (UAV) [26,27], and smart car applications [28], as well as their gradual use in the hard environment of underground coal mines [29,30], show the technology’s great environmental resilience.

We are devoted to giving coal mining machines extensive and long-term autonomous positioning and orientation capabilities. The following contributions are ours:

We analyzed inertial measurement unit (IMU) calibration and UWB positioning errors. We also performed a simulation analysis on the optimal spatial layout of base stations for IMU-assisted UWB positioning systems. These analyses lay a solid theoretical foundation for subsequent integrated navigation models.

In order to enable long-term autonomous navigation and positioning of coal miners in mines, we use the ESKF to filter the errors in the measurement data from IMU-assisted UWB positioning to obtain an optimal estimate of the error for the current situation and to correct for inaccuracies due to approximations.

We have also improved the combined positioning model in terms of both adaptive estimates and non-visual processing. Using actual data, the findings are now within 10 cm.

## 2. IMU-Assisted UWB Positioning System Architecture Analysis

The IMU-assisted UWB positioning system consists of the shearer with a built-in high-precision six-axis IMU tag, four base stations, and a PC-based host computer that receives and processes the data. The IMU-assisted UWB positioning system data tight coupling method is shown in Figure 1.

### 2.1. The Principle of UWB Ranging

#### 2.1.1. Single-Sided Two-Way Ranging

Single-sided Two-Way Ranging (SS-TWR) involves simply measuring the round-trip delay of a single message from one base station to another and the response back to the original base station.

As shown in Figure 2, the tag T0 sends a Poll to base station A0. Upon receiving the Poll, base station A0 responds with an R. The process ends when tag T0 receives the R. $T_{round}$ is the time it takes for tag T0 to send a message and receive a response. $T_{reply}$ is the time it takes for base station A0 to receive the Poll and send an R in response. The time required for tag T0 to reach base station A0 is:(1)$$T_{prop}=\frac{T_{round}-T_{reply}}{2}$$

Because tag T0 and base station A0 use their respective local clocks for counting, their clock offset errors are different. Therefore, the measured flight time will have an error, which increases as the counting time increases. The speed of light is very fast, and even a small time error can result in a significant distance error. Therefore, this method cannot be used for ranging.

#### 2.1.2. Double-Sided Two-Way Ranging

The UWB positioning system uses an optimized message exchange scheme in which the tag sends a broadcast message, a single polling message received by all base stations, and a response message from each base station in turn, after which the tag completes the range exchange by sending the resultant message received by all base stations. Each device is precisely marked with the transmission and reception times of the messages. Figure 3 shows a case of a tag message at a base station, labeled with tag T0 reply time $T_{reply1}$ and round trip time $T_{round1}$ and base station A0 reply time $T_{reply2}$ and round trip time $T_{round2}$. Then, each base station receiving the resultant message has enough information to calculate the time of flight (TOF) between itself and the tag. In Figure 3, the TOF value is labeled as $T_{\mathrm{prop}}$, which indicates the propagation time of the message between the tag and each base station. The ranging scheme has very good performance in that it measures the round-trip time from both sides and can be used to compensate for the clock offset between the base station and the tag. From the SS-TWR method, we can obtain:(2)$$T_{\mathrm{prop}\;}=\frac{1}{2}\left(T_{\mathrm{round}1}-T_{\mathrm{reply}1\;}\right)$$
(3)$$T_{\mathrm{prop}\;}=\frac{1}{2}\left(T_{\mathrm{round}2}-T_{\mathrm{reply}2\;}\right)$$
(4)$$\begin{array}{c} T_{\mathrm{round}1}\times T_{\mathrm{round}2}=\left(2T_{\mathrm{prop}\;}+T_{\mathrm{reply}1}\right)\left(2T_{\mathrm{prop}\;}+T_{\mathrm{reply}2}\right)\\ =4T_{\mathrm{prop}\;}^{2}+2T_{\mathrm{prop}\;}\left(T_{\mathrm{reply}1}+T_{\mathrm{reply}2}\right)+T_{\mathrm{reply}1}T_{\mathrm{reply}2} \end{array}$$
(5)$$\begin{array}{c} T_{\mathrm{round}1}\times T_{\mathrm{round}2}-T_{\mathrm{reply}1}T_{\mathrm{reply}2}=4T_{\mathrm{prop}\;}^{2}+2T_{\mathrm{prop}\;}\left(T_{\mathrm{reply}1}+T_{\mathrm{reply}2}\right)\\ =T_{\mathrm{prop}\;}4T_{\mathrm{prop}\;}+2T_{\mathrm{reply}1\;}+2T_{\mathrm{reply}2\;}\\ =T_{\mathrm{prop}\;}\left(T_{\mathrm{round}1}+T_{\mathrm{round}2}+T_{\mathrm{reply}1}+T_{\mathrm{reply}2}\right) \end{array}$$

Thus, we can calculate the formula for $\;T_{\mathrm{prop}\;}$:(6)$$T_{prop}=\frac{T_{round1}\times T_{round2}-T_{reply1}\times T_{reply2}}{T_{round1}+T_{round2}+T_{reply1}+T_{reply2}}$$

The errors introduced by the clock using the UWB bilateral ranging method are as follows:(7)$$error=T_{prop}\times \left(1-\frac{K_{A0}+K_{T0}}{2}\right)$$
where $K_{A0}$ and $K_{T0}$ are the ratios of the actual and expected frequencies of the base station A0 and tag T0 clocks, respectively. Therefore, the error of distance measurement caused by the bilateral ranging method is negligible.

### 2.2. UWB Positioning Principle

Precision positioning, strong anti-interference performance, high transmission rate, huge system capacity, low transmission power, low power consumption, good security, and minimal system complexity are all benefits of UWB communication technology. It is ideal for applications requiring high-precision positioning in challenging settings.

This section may be divided into subheadings. It should provide a concise and precise description of the experimental results, their interpretation, as well as the experimental conclusions that can be drawn.

The UWB positioning system is managed by using time division multiple access to enable each tag to communicate with the base station to obtain the distance value d between the tag and the base station, where the task of managing the time slice is handled by the base station A0.
(8)$$d=C\times T_{propA}\;,C=3*10^{8}\;m/s$$

Using a trilateral ranging positioning method in the same reference coordinate system, where the coordinate values of all base stations are known, the UWB positioning system deploys N base stations in the mining area. Figure 4 shows the schematic diagram of the classical trilateral ranging method for UWB positioning. Record the coordinate values of the *i* base station as ($x_{i},\;y_{i},\;z_{i}$) (*i* = 1, 2, ⋯, *n*). Let the coordinate distance value of the unknown point label $T_{n}$ to the *i*th base station be $d_{i}$ (*i* = 1, 2, ⋯, *n*). Drawing a circle with $d_{1}$, $d_{2}$, ⋯, $d_{i}$ as the radius among them, the focal point, and the position of the unknown point are calculated according to the Pythagorean theorem as follows:(9)$$\left\{\begin{array}{c} {\left(x-x_{1}\right)}^{2}+{\left(y-y_{1}\right)}^{2}+{\left(z-z_{1}\right)}^{2}=d_{1}^{2}\\ {\left(x-x_{2}\right)}^{2}+{\left(y-y_{2}\right)}^{2}+{\left(z-z_{2}\right)}^{2}=d_{2}^{2}\\ \cdots \\ {\left(x-x_{n}\right)}^{2}+{\left(y-y_{n}\right)}^{2}+{\left(z-z_{n}\right)}^{2}=d_{n}^{2} \end{array}\right.$$

The system of equations is eliminated to obtain a ternary N − 1 linear system of equations as follows:(10)$$2\left[\begin{array}{ccc} x_{1}-x_{n} & y_{1}-y_{n} & z_{1}-z_{n}\\ x_{2}-x_{n} & y_{2}-y_{n} & z_{2}-z_{n}\\ \cdots & & \\ x_{n-1}-x_{n} & y_{n-1}-y_{n} & z_{n-1}-z_{n} \end{array}\right]\left[\begin{array}{c} x\\ y\\ z \end{array}\right]=-\left[\begin{array}{c} d_{1}^{2}-d_{n}^{2}\\ d_{2}^{2}-d_{n}^{2}\\ \ldots \\ d_{n-1}^{2}-d_{n}^{2} \end{array}\right]+c$$
where *c* denotes the constant associated with the coordinates of the base station. This constant represents a specific value or coefficient used in the calculation or representation of the base station’s location. The expression of the above ternary N − 1 linear system of equations is shaped as AX = B. Using the minimum mean square error estimate, the coal miner position coordinates are obtained as:(11)$$\left[\begin{array}{c} x\\ y\\ z \end{array}\right]=-\frac{1}{2}{\left(A^{T}A\right)}^{-1}A^{T}\left(\left[\begin{array}{c} d_{1}^{2}-d_{n}^{2}\\ d_{2}^{2}-d_{n}^{2}\\ \cdots \\ d_{n-1}^{2}-d_{n}^{2} \end{array}\right]-c\right)$$

Thus, the coordinate position (*x*, *y*, *z*) of the label is solved.

### 2.3. UWB Positioning System Error Analysis

The inherent property matrix A of the UWB positioning system is the difference in coordinates of each base station. When there is an error in the coordinate distance value $d_{i}($*i* = 1, 2,⋯, *n*) from the unknown point label $T_{n}$ to the first base station, the measurement error will be accumulated to the position estimation by the operation.
(12)$$A=\left[\begin{array}{ccc} x_{1}-x_{n} & y_{1}-y_{n} & z_{1}-z_{n}\\ x_{2}-x_{n} & y_{2}-y_{n} & z_{2}-z_{n}\\ \cdots & & \\ x_{n-1}-x_{n} & y_{n-1}-y_{n} & z_{n-1}-z_{n} \end{array}\right]$$
where let $M=-0.5{\left(A^{T}A\right)}^{-1}A^{T}$, the cumulative error maps the measurement error Δb to the ${\;[\Delta x\;\Delta y\;\Delta z]}^{T}$ error space through the matrix M. For a matrix M with singular value decomposition $M=UDV^{T}$, the rank of the matrix rank (M) = K. U and *V* is a standard orthogonal matrix, in terms of error transfer as follows:(13)$$\begin{array}{c} \Delta x=M\Delta b=\left[\delta_{1}u_{1},\delta_{2}u_{2},\ldots ,\delta_{k}u_{k}\right]\left[\begin{array}{c} v_{1}^{T}\\ v_{2}^{T}\\ \ldots \\ v_{k}^{T} \end{array}\right]\Delta b\\ \;\;\;\;\;\;\;\;\;\;\;\;\;\;\;=\left[\delta_{1}u_{1},\delta_{2}u_{2},\ldots ,\delta_{k}u_{k}\right]\left[\begin{array}{c} y_{1}\\ y_{2}\\ \ldots \\ y_{k} \end{array}\right] \end{array}$$

The vector $y={\left[\begin{array}{c} y_{1}\;\;\;\;y_{2}\;\;\;\;\cdots \;\;\;\;y_{k} \end{array}\right]}^{T}=V^{T}\Delta b$ is obtained by rotating through the matrix $V^{T}$. In the given formula, δ (delta) represents a vector. Specifically, it is a vector with *k* elements, denoted as $\delta =\left[\delta_{1},\delta_{2},\cdots ,\delta_{n}\right]$. Each $\delta_{i}$ represents the *i* element in the vector. In this formula, δ is obtained as the result of a matrix-vector multiplication.

If ∥Δb∥=1 denotes the unit error sphere in n − 1 dimensional space, the sphere becomes an ellipsoid, as shown in Figure 5 after mapping by matrix M. The ellipsoidal plane represents the position error space of the system, and the magnitude of the singular values of the matrix M determines its axis length or stretching or compression. Thus, the matrix M reflects the spatial layout of the base station and also determines the intrinsic characteristics of the system. The optimal matrix M is obtained when its singular value is small enough so that the system can act as an attenuation when the proper measurement value has a large error.

Thus, an unreasonable layout leads to an undesirable localization solution, whose minimum mean square error estimate only represents an optimal estimate in an algebraic sense and does not guarantee that the estimate is optimal in a geometric sense.

### 2.4. UWB Positioning System Base Station Layout

The optimal spatial layout of the base station should not only consider the performance of the positioning system accuracy but also ensure that the volume of space that the base station can cover is large enough. Singular Value Decomposition (SVD) and Dilution of Precision (DOP) are two different approaches used for error analysis in positioning systems. Here is a brief comparison between the two:

The SVD-based error analysis scheme can assess the placement of base stations, evaluate the accuracy of distance estimates, and provide insights into the overall system performance. It helps identify potential error sources and can aid in system calibration and improvement.

The DOP analysis is commonly used in GPS and UWB applications to evaluate the quality of the satellite or base station configuration. It assists in understanding the reliability of position measurements by considering the geometric distribution of the satellites or base stations.

In summary, SVD focuses on analyzing the least squares solution and providing insights into the quality and uncertainties of estimated positions, while DOP analysis assesses the geometric configuration of satellites or base stations to evaluate the positioning system’s accuracy and precision.

The evaluation index of the base station layout is expressed in mathematical language as follows:(14)bSi…bSi=FxbSi…bSnargmin=meanδ+varδ−λVbSi…bSnbSi…bSnargmin

The surrogate function *F*(*x*) is the difference between the mean of the singular values, the variance, and the coverage volume of the base station of the matrix M.

Where $\delta =SVD(-0.5{\left(A^{T}A\right)}^{-1}A^{T})$ denotes the singular value vector of the matrix M for SVD decomposition. $V(bS_{i}\ldots bS_{n})$ denotes the volume of the polyhedron with the base station as the vertex. $bS_{i}=constrant$ indicates the space constraint. If the base stations are mostly installed in areas such as walls and corners, *λ* is a weighting parameter indicating the trade-off between spatial coverage volume and positioning performance of the positioning system. In this work, the IMU-assisted UWB positioning system is modeled and simulated to compare the impact of the base station on the positioning accuracy of the system under different layouts. The simulated scenario changes the height of the base station A0 node to produce three different spatial layouts, as shown in Figure 6. The red dots represent base stations, and the blue triangles represent tags.

Calculate the sizes of the error ellipsoids for the three layout schemes based on the theoretical knowledge from Section 2.3 and summarize the lengths of the error ellipsoid axes in Table 1 as follows. Gradually increasing the length of the system space on the height axis from Layout 1 to Layout 3, the length of a certain axis of the ellipsoid gradually becomes smaller, which shows the more effective the weakening of the error.

In a practical scenario, the base station layout can easily ensure a long enough coverage distance in the horizontal plane. However, in the height axis, due to the limited coverage of factors such as the height under the mine, erection costs, etc., the height axis too low spatial discrimination leads to a larger calculation error in the Z-axis than the X-axis Y-axis. This indicates that the spatial width of the base station layout determines the resolution of the positioning system. 

### 2.5. IMU System Error Analysis

Inertial components vary greatly in performance depending on the manufacturing process, materials, and cost, ranging from navigation grade, tactical grade, industrial grade to commercial grade. The focus of this article is on low-cost MEMS inertial navigation systems, aiming to provide a theoretical reference for MEMS inertial measurement unit applications. Inertial elements generally refer to accelerometers and gyroscopes, which are capable of sensing linear acceleration and angular velocity of rotation in three orthogonal directions. In general, the main sources of errors affecting inertial navigation performance include inertial element accuracy and system errors, such as alignment errors of orthogonal axes, sensor scale factors, non-orthogonality, and random noise, among others.

The IMU is left stationary for a period of time, and the magnitude of the inertial navigation position drift is analyzed as time grows. Of course, the position estimation assumes that the navigation error is caused by accelerometer bias only. In practice errors, in position estimation are caused by many factors, including gyroscope offset stability, accelerometer scale factor uncertainty, and other factors, but the main contributing factor is caused by uncorrected accelerometer bias. Any deviation in the horizontal axis of the accelerometer can lead to errors in the estimation of the object’s attitude. The error in the attitude matrix is directly related to the difference between the acceleration value and the actual gravity value under the coordinate transformation to the same reference system. The motion acceleration deviation is transferred to the position error by integration so that the position error due to accelerometer deviation will grow as a square with time.

Therefore, the interference of seemingly small errors in the inertial elements on the navigation output cannot be ignored.

All deterministic errors of the inertial measurement unit are collated and expressed uniformly in the equation, resulting in the following error correction model as follows:(15)$$\bar{a}=Ca+b+w$$
where *a* denotes the true 3 × 1 vector of inertia values. $\bar{a}$ indicates the vector of 3 × 1 inertia measurements to be corrected. *C* is the scale factor and the unaligned error correction matrix. *b* indicates the static drift of the inertial measurement unit. w indicates the noise of the inertial measurement unit. The specific equation is as follows:(16)$$\left[\begin{array}{c} {\bar{a}}_{x}\\ {\bar{a}}_{y}\\ {\bar{a}}_{z} \end{array}\right]=\left[\begin{array}{c} C_{11}\;\;\;\;C_{12}\;\;\;\;C_{13}\\ C_{21}\;\;\;\;C_{22}\;\;\;\;C_{23}\\ C_{31}\;\;\;\;C_{32}\;\;\;\;C_{33} \end{array}\right]\left[\begin{array}{c} a_{x}\\ a_{y}\\ a_{z} \end{array}\right]+\left[\begin{array}{c} b_{ax}\\ b_{ay}\\ b_{az} \end{array}\right]+w$$

To estimate the magnitudes of parameters *C* and *b*, an effective estimation algorithm involves obtaining a sufficient number of data pairs through sampling and performing minimum mean square error estimation on them. Clearly, this is easily achievable by simply allowing the sensors to remain stationary in various orientations for a brief period and taking measurements, thereby obtaining a sufficient number of samples. Due to the difficulty in parameter estimation for *C* and *b* in the calibration model, the unknown variables are separated from the known variables, and the equation is rewritten as:(17)$$\left[\begin{array}{c} {\bar{a}}_{x}\\ {\bar{a}}_{y}\\ {\bar{a}}_{z} \end{array}\right]=\left[\begin{array}{cccccccccc} a_{x} & a_{y} & a_{z} & & & & & & & \\ & & & a_{x} & a_{y} & a_{z} & & & & |I_{3\times 3}\\ & & & & & & a_{x} & a_{y} & a_{z} & \end{array}\right]\left[\begin{array}{c} C_{1*}^{T}\\ C_{2*}^{T}\\ C_{3*}^{T}\\ b \end{array}\right]+w_{noise}$$
where $C_{k*}^{T}$ denotes the transpose of all elements in the *k* row of matrix *C*. Therefore, the measurement principle of inertial data can be expressed as an equation of the form $y=hx+w_{noise}$, where x represents the parameters to be estimated, h and y denote the input and output data, respectively. The problem is then transformed into parameter fitting for a linear equation.

To ensure that the sample data are globally representative, the sampled measurements of the forces should cover as much of the sample space as possible. Simply, one axis of the fixed inertial element is perpendicular to the direction of gravity and is rotated about 45° around this axis in turn so that gravity appears in various force decompositions in the other two directional axes with positive and negative combinations. Similarly, this operation was repeated by replacing the fixed axis to obtain 24 sets of sample data. Figure 7 below shows the inertial component pose placement.

Bringing all the above data into the equation yields a system of linear equations:(18)$$\left\{\begin{array}{c} y_{1}=h_{1}x\\ y_{2}=h_{2}x\\ \cdots \\ y_{24}=h_{24}x \end{array}\Rightarrow \left[\begin{array}{c} y_{1}\\ y_{2}\\ \cdots \\ \cdots \\ y_{23}\\ y_{24} \end{array}\right]=\left[\begin{array}{c} h_{1}\\ h_{2}\\ \cdots \\ \ldots \\ h_{23}\\ h_{24} \end{array}\right]x+w_{\mathrm{noise}}\right.$$

That is, $y=Hx+w_{noise}$, using the minimum mean square error to estimate the parameter x, gives:(19)$$\hat{x}={\left(H^{T}R^{-1}H\right)}^{-1}H^{T}R^{-1}y$$
where R is the noise covariance matrix of the three-axis accelerometer. The final accelerometer calibration equation is, therefore, as follows:(20)$$a=C^{-1}\left(\overset{‾}{a}-b\right)$$

## 3. Combined Positioning System Model

Similar to EKF, ESKF is a nonlinear filter for time-varying systems. However, unlike the EKF, it is always linearized around 0, so the linearization is more accurate. It is not filtering the navigation information but filtering the errors in the navigation information. Because the error is a small amount, the linearization is more accurate due to the approximations made when doing the IMU solving, and those rounded off give errors to the IMU solving. What the ESKF does is filter the error to obtain the best estimate of the error for the current situation and then subtract this error from the calculated displacement, velocity, attitude, and other quantities to correct for inaccuracies due to approximations. Finally, the IMU-assisted UWB localization system is optimized in terms of both non-visual range processing and adaptive estimation.

### 3.1. System Coordinate System

This UWB positioning system selects the forward direction of the carrier under the mine as the positive *x*-axis direction, the right flank direction as the positive *y*-axis direction, and the downward direction as the positive *z*-axis direction. Figure 8 shows the schematic diagram of the rotation direction of the MEMS IMU. Therefore, the carrier coordinate system chosen for this article is based on this standard, and the world coordinate system is chosen directly from the ENU coordinate system.

### 3.2. Equation of State

In filters, the equation of state is generally written in the following form:(21)$$\dot{X}=F_{t}X+B_{t}W$$
where X is the state quantity we want to estimate. For IMU-assisted UWB data fusion localization systems, there are various options for the estimation of the state. In this article, the displacement error, velocity error, attitude error, bias error of the gyroscope, and bias error of the accelerometer are selected and written as vectors as follows:(22)$$X={\left[\begin{array}{c} \delta P^{T}\;\delta V^{T}\;\delta \phi^{T}\;\varepsilon^{T}\nabla^{T} \end{array}\right]}^{T}$$
where the displacement error is $\delta P^{T}={\;[\delta P_{E}\;\delta P_{N}\;\delta P_{U}]}^{T}$, Velocity error is $\delta V^{T}={\;[\begin{array}{ccc} \delta V_{E} & \delta V_{N} & \delta V_{U} \end{array}]}^{T}$, Attitude error is $\delta \phi^{T}={\;[\begin{array}{ccc} \delta \phi_{E} & \delta \phi_{N} & \delta \phi_{U} \end{array}]}^{T}$, The bias error of the gyroscope is $\varepsilon^{T}={\;[\begin{array}{c} \delta \varepsilon_{x\;}\varepsilon_{y}{\;\varepsilon }_{z} \end{array}]}^{T}$, The bias error of the accelerometer is $\nabla^{T}={\;[\begin{array}{ccc} \nabla_{x} & \nabla_{y} & \nabla_{z} \end{array}]}^{T}$.

The bias noise of the gyroscope and the bias noise of the accelerometer in the IMU, their complete forms are as follows:(23)$$W={\;[\begin{array}{cccccc} \omega_{gx} & \omega_{gy} & \omega_{gz} & \omega_{ax} & \omega_{ay} & \omega_{az} \end{array}]}^{T}$$

The first three of these are the bias noise of the gyroscope in three axes and the last three are the bias noise of the accelerometer in three axes. $C_{b}^{n}$ is the transformation of the IMU sensor to the navigation system n at the current moment t.

The noise transfer matrix $B_{t}$ is formulated as follows:(24)$$B_{t}=\left[\begin{array}{c} \begin{array}{cc} 0_{3\times 3} & 0_{3\times 3}\\ 0_{3\times 3} & C_{b}^{n} \end{array}\\ \begin{array}{cc} {-C}_{b}^{n} & 0_{3\times 3}\\ 0_{6\times 3} & 0_{6\times 3} \end{array} \end{array}\right]$$

The complete $F_{t}$ matrix is as follows:(25)$$F_{t}=\left[\begin{array}{ccccc} 0_{3\times 3} & I_{3\times 3} & 0_{3\times 3} & 0_{3\times 3} & 0_{3\times 3}\\ 0_{3\times 3} & 0_{3\times 3} & F_{23} & 0_{3\times 3} & C_{b}^{n}\\ 0_{3\times 3} & 0_{3\times 3} & F_{33} & -C_{b}^{n} & 0_{3\times 3}\\ 0_{3\times 3} & 0_{3\times 3} & 0_{3\times 3} & 0_{3\times 3} & 0_{3\times 3}\\ 0_{3\times 3} & 0_{3\times 3} & 0_{3\times 3} & 0_{3\times 3} & 0_{3\times 3} \end{array}\right]$$
(26)$$F_{23}=\left[\begin{array}{ccc} 0 & -f_{U} & f_{N}\\ f_{U} & 0 & -f_{E}\\ -f_{N} & f_{E} & 0 \end{array}\right]$$
(27)$$F_{33}=\left[\begin{array}{ccc} 0 & \omega \sin L & -\omega \cos L\\ -\omega \sin L & 0 & 0\\ \omega \cos L & 0 & 0 \end{array}\right]$$

$f^{n}={\left[\begin{array}{c} f_{E}\;\;\;\;f_{N}\;\;\;\;f_{U} \end{array}\right]}^{T}$ represents the representation of the measured acceleration in the navigation frame using an IMU. $\omega_{ie}^{n}=[0,\omega \cos L,\omega \sin L{]}^{T}$ represents the angular velocity vector relative to the Earth coordinate system, expressed in the n-frame. Here, ω denotes the magnitude of the Earth’s rotational rate, approximately $7.292115\times 10^{-5}$ rad/s, and L represents the latitude.

### 3.3. Observation Equation

The predicted values mentioned before are from IMU, while for the observed values, we use the UWB observations. In the filter, the observation equation takes the following form:(28)$$Y=G_{t}X+C_{t}N$$
(29)$$Y={\left[\begin{array}{c} \delta P_{E}\;\delta P_{N}\;\delta P_{U} \end{array}\right]}^{T}$$

δP represents the displacement error in the East, North, and Up directions, respectively. The subscript E stands for East, N stands for North, and U stands for Up. From the form of X-state quantities, it can be deduced that $G_{t}=\left[\begin{array}{cc} I_{3\times 3} & 0_{3\times 12} \end{array}\right],\;\;C_{t}=[\begin{array}{c} I_{3\times 3} \end{array}]$. *N* in the observation equation is the observation noise, which is due to the non-line-of-sight effects received by the UWB base station, and it has the following complete form:(30)$$N=\left[\begin{array}{ccc} n_{P_{E}} & n_{P_{N}} & n_{P_{U}} \end{array}\right]$$

### 3.4. ESKF Data Fusion

The IMU-assisted UWB data fusion positioning system equation of state and observation equations are obtained in the above two sections. To be applied to the filter, it must be discretized, and the discretization is carried out according to the sampling time. For $F_{t}$ in equation. using the first-order Taylor approximation yields: $F_{k-1}=I_{15\times 15}+F_{t}({\hat{X}}_{k-1},v_{k},0)T$, where T is the Kalman filtering period. $v_{k}$ represents the process noise at time step k. In the context of the equation, $v_{k}$ is used as an input to the function $F_{t}$, which represents the state transition matrix or the Jacobian of the system dynamics. The function $F_{t}$ takes as input the previous state estimate ${\hat{X}}_{k-1}$, process noise $v_{k}$, and the period T, and generates a matrix that approximates the state transition between time steps (k − 1) and k. The final ESKF data fusion is obtained. Formulas (31) and (32) describe the prediction process, which is mainly used to predict state quantities and the corresponding covariance of the state quantities. Formula (33) $K_{k}$ refers to the Kalman gain, which is used to determine the weighting between the prediction and measurement in the current iteration, determining which one should be trusted more. Equations (34) and (35) are used to correct the errors in the predictions made by the previous two equations.
(31)$${\overset{ˇ}{X}}_{k}=F_{k-1}{\hat{X}}_{k-1}+B_{k-1}W_{k}$$
(32)$${\overset{ˇ}{P}}_{k}=F_{k-1}{\hat{P}}_{k-1}F_{k-1}^{T}+B_{k-1}Q_{k}B_{k-1}^{T}$$
(33)$$K_{k}={\overset{ˇ}{P}}_{k}G_{k}^{T}{\left(G_{k}{\overset{ˇ}{P}}_{K}G_{k}^{T}+C_{k}R_{k}C_{k}^{T}\right)}^{-1}$$
(34)$${\hat{P}}_{k}=\left(I-K_{k}G_{k}\right){\overset{ˇ}{P}}_{k}$$
(35)$${\hat{X}}_{k}=X_{k}+K_{k}\left(Y_{k}-G_{k}{\overset{ˇ}{X}}_{k}\right)$$

### 3.5. Combined Positioning Model Optimization Processing

#### 3.5.1. Non-Line-of-Sight Processing

There is a very significant jitter error in UWB positioning systems, known as non-visual range error. Non-visual range errors are closely related to the current environment and are prone to extremely inaccurate range values when there are obstacles in the linear path of the transmitted signal or when multipath effects are evident. The root cause of range inaccuracy is that the UWB hardware accepts multipath electromagnetic signals and recognizes the first signal reached as a straight signal with no delay, a mechanism that performs well for most line-of-sight situations. The signal power attenuation on the non-line-of-sight path of the receiving signal is larger than that of the straight path signal, so this feature can be used to determine the effectiveness of the UWB hardware straight path signal error detection. The specific method of non-visual distance judgment is as follows:(36)$$dB=fP-rxP=\left\{\begin{array}{c} <Threshold1\\ \geq Threshold2\; \end{array}\right.\begin{array}{c} \;\;LOS\\ \;\;NLOS \end{array}$$
where fP denotes the first path power of the wireless signal accepted by the UWB, rxP represents the received power at the receiver, and dB represents signal gain, i.e., the ratio of the first path power to the total wireless power signal. A small dB value indicates that the total power of the wireless signal is mainly concentrated in the first path, and the ranging quality is good. On the contrary, the power of the signal is scattered in various paths, and the ranging quality is poor, which is most likely to be a non-visual situation. In addition, the tightly coupled combined navigation model provides a probabilistic consistency test for data plausibility.

In ESKF, the innovation is an orthogonal, temporally uncorrelated, white noise sequence, a normalized Gaussian noise.
(37)$$q\left(i\right)=v^{T}\left(i\right)S\left(i\right)v\left(i\right)\subset \chi^{2}$$

The equation $v\left(i\right)=r_{i}-||bS_{i}||$ represents the innovation residual of the observed data, where S(i) is the innovation covariance matrix. The random variable q(i), referred to as the normalized innovation variance, follows a chi-squared $\chi^{2}$ distribution with degrees of freedom m=dim(z(k)), a mean of m, and a variance of 2 m. $r_{i}$ represents the residual or the difference between the observed measurement z(k) and its predicted value h(k). z(k) represent the measured coordinates or distances obtained from UWB sensors. The chi-square test provides a confidence bound for verifying the external observables and a confidence estimate for the hypothesis H: {whether q(i) is a valid value} based on the chi-square distribution property.

If the hypothesis H: {the current observation is not an outlier} exists, the probability that the hypothesis will be accepted is as follows:(38)$$P\left\{Haccepted|q_{i}\in \left(2-\sigma \mathrm{gate}\right)\right\}$$

According to the properties of the chi-squared distribution, the ±σ and ±2σ boundaries for the innovation sequence can be determined. The ±2σ boundary provides a 95% confidence level, indicating that approximately 95% of the innovations fall within the 2−σ range.

The above two methods give the theoretical basis for non-visual detection in terms of physical properties and probabilistic models, respectively, which work in conjunction with each other. The integration navigation model, utilizing outlier processing techniques, can effectively reduce the interference of non-line-of-sight on the positioning system. When the system encounters a non-line-of-sight situation, it automatically switches to inertial navigation for position estimation, which results in smoother state trajectory curves and enhanced stability.

#### 3.5.2. Adaptive Estimation

In the complex environment of underground coal mines, the probabilistic statistical properties of noise are not invariable. For example, the inertial measurement unit is susceptible to temperature and humidity changes and shifts in measured values. UWB ranging is subject to multipath interference errors in some environments. Although it is not possible to accurately model this random noise, an effective and feasible way is to properly adjust the mean and variance of the noise so that the model approximates white noise more closely to the real situation. The adaptive estimation of the ESKF filtering algorithm lies in the ability to identify the mean and variance of the noise online to adapt to the complex and variable noise characteristics. In general, the amplitude of the inertial data noise of the inertial measurement unit varies slowly, and to simplify the computational process, adaptive estimation only considers the mean and variance of the online recognition observation noise. In a way, the innovation value reflects the performance of the Kalman filter estimation, so Sage–Husa adaptive estimation uses this property for the online identification of process and measurement noise. Let r be the deviation value of the measured noise, then the online identification of the noise characteristics is as follows:(39)$$r_{k+1}=\left(1-d_{k+1}\right)r_{k}+d_{k+1}\left(\delta z_{k}-h\left(\delta x_{k}^{-}\right)\right)$$
(40)$$R_{k+1}=\left(1-d_{k+1}\right)R_{k}+d_{k}\left(\xi_{k+1}\xi_{k+1}^{T}-H_{k+1}P_{k+1}^{-}H_{k+1}^{T}\right)$$
(41)$$\xi_{k+1}=Z_{k+1}-H_{k+1}X_{k+1^{\cdot }}-r_{k+1}$$
where $d_{k+1}$ denotes the weight of the current residual, and b is the decay factor, taking values in the range of 0.95–0.99. The mean $r_{k+1}$ and variance $R_{k+1}$ estimates of the measurement noise are expressed as a sequence of historical residuals $\left\{r_{1}r_{2}\ldots r_{k}\right\}$ weighted with the new residuals, and the historical residual information will contribute less and less to the discrimination algorithms as time increases. The variables $\delta z_{k}$ and $\delta x_{k}^{-}$ represent the increments or changes in the measurement and state vectors, respectively, at time step *k*. Sage–Husa adaptive estimation, a suboptimal estimator that uses the autocorrelation property of the residual series to count the mean and variance of the system noise online, has good results in practical applications.

## 4. Experiments

### 4.1. Experimental Platform

The experimental integrated mining working face consists of a coal wall, upper and lower chutes, quarry space, and mechanical equipment related to the integrated mining process. The coal wall is 8.14 × 3.88 × 2.2 m, the working face space length is 12.94 × 2.2 m, the minimum/large roof control distance is 3.6/4.0 m, and the roof is supported by 16 hydraulic supports in total. Among them, four sets of support cover type brackets, four sets of cover type brackets, four sets of top coal release hydraulic brackets, two sets of support type hydraulic brackets, and two sets of end hydraulic brackets. Main equipment: Double drum pin rail type seamless traction coal mining machine, medium double chain end discharge scraper conveyor, medium double chain scraper transfer machine, etc.

Build a hardware-working platform for UWB positioning and inertial navigation. The platform includes four UWB base stations, UWB tags with built-in high-precision six-axis IMU fixed on the coal mining machine, MEMS inertial measurement unit, and Raspberry Pi as the main controller. Among the UWB device DW1000 high precision positioning chip, the inertial measurement unit selected low-cost six-axis ICM-42605 inertial components. The experimental platform of the header mining simulation working face is shown in Figure 9.

### 4.2. Simulation Experiment of Header Working Face

The UWB tag with built-in high-precision six-axis IMU is connected to the Raspberry Pi as the master controller and fixed to the coal mining machine. Through Raspberry Pi, we collect the serial data of the tag. The serial data includes the current distance value of the tag to the four base stations as well as the acceleration value of the three axes and the gyroscope value of the three axes to get the real-time serial data.

Simulation experiments, such as the calculation of the algorithm, are carried out by MATLAB, and the final trajectory map of fusion localization is obtained in Figure 10. The red triangle indicates the location of the base station, the pink dot indicates the trajectory calculation of pure UWB measurement data, and the blue line indicates the trajectory of IMU-assisted UWB data fusion. Where Figure 10b shows a partially enlarged view of Figure 10a.

UWB positioning errors are stable over long periods, while inertial navigation positioning errors are accurate over short periods. A single navigation system cannot achieve precise positioning and navigation, and there are features such as complementary advantages between different positioning systems. The combined positioning model effectively addresses NLOS interference or occlusion in GPS denial scenarios. When there are NLOS errors at the base station, the combined positioning model switches the positioning solution to the inertial navigation system in time and then switches back to the combined model when the NLOS interference disappears. The combined positioning system, therefore, has the advantages of high accuracy, stability, and adaptability to the environment. Figure 11 indicates that the UWB data is lost, and the IMU can work continuously. Where Figure 11b shows a partially enlarged view of Figure 11a.

The article implements a localization system that integrates data from IMU and UWB sensors using ESKF. It also includes a positioning system based solely on ranging UWB measurements, known as Only-Ranging UWB. As shown in Figure 12, the error curves of the position outputs for the two localization systems are plotted. From Figure 12, it can be observed that when there is NLOS (Non-Line-of-Sight) error in the ranging measurements, the output of the Only-Ranging UWB system becomes unstable, resulting in significant positional jitter. On the other hand, the performance of the integrated positioning system remains unaffected, with errors effectively controlled within 0.1 m. This article addresses the vulnerability of the Only-Ranging UWB positioning system to NLOS interference. Additionally, it can be observed that the positioning system exhibits poor performance in the vertical direction, which validates the conclusion of the system error propagation due to base station deployment discussed in Section 2.4.

As shown in Figure 13, the green line indicates the real trajectory. The blue line indicates the trajectory of the UWB positioning system. The red line indicates the trajectory using the ESKF to fuse the data from both IMU and UWB sensors. They are plotted at each output position trajectory. As can be seen in Figure 13, the trajectory on the *XYZ* axis from the fusion of the two sensor data, IMU and UWB, using the ESKF fits better with the real trajectory. Whereas the trajectory of the UWB positioning system has a greater impact on the trajectory on the *XY* axis, the trajectory on the *Z* axis has a greater trajectory error due to the low spatial resolution.

A block diagram of the IMU-assisted UWB positioning system flow is shown in Figure 14 below. The blue axis in Figure 14 represents the main line of the combined positioning model, which implements the inertial navigation system, performs position nudging, and outputs three-axis position, velocity, and attitude in real-time. The purple axis shows the calibration solution, which implements Sage–Husa adaptive filtering, NLOS discrimination, etc., and gives the current state error of the system. The red axis indicates the filtering feedback, which feeds back the amount of error in the measurement update solution to the system state.

### 4.3. Analysis of Experimental Results

The effect of IMU-assisted UWB data fusion positioning system navigation is obvious compared to single navigation, especially when the low vertical spatial resolution cannot be resolved on the UWB layout, and the fusion algorithm is effective. With the current use of the actual data, the effect can also be controlled within 10 cm, and the positioning accuracy can be improved, which has proved that the filtering method is reliable and effective. It was also demonstrated that where UWB positioning is subject to multipath effects and non-line-of-sight (NLOS) for short periods resulting in data loss, etc., accurate inertial navigation positioning errors for short periods will continue to work.

Due to the limited experimental conditions, experiments with a UWB positioning system with more base stations and over a larger area were not carried out. Future research will focus on real-life coal mine experiments using more base stations to further demonstrate the effectiveness of the proposed approach. While the small number of UWB base stations and the limited operating range has the potential to improve the performance of the positioning system, the trade-off between positioning accuracy and the cost of base station deployment remains a problem to be solved. In conclusion, for coal miners working in a limited area, the proposed algorithm can basically meet the positioning accuracy requirements in GPS rejection scenarios.

## 5. Conclusions

We investigated the problem of locating and orientating shearers in a GPS-denied coal mine environment. To solve this problem, we propose an IMU-assisted UWB data fusion-based system for shearer location in underground coal mines and fuse the two sensor data using ESKF. This uses data from the IMU sensor displacement error, velocity error, attitude error, gyroscope bias error, and accelerometer bias error to establish the filter state equations and the UWB sensor measurements as the filter observation equations. Instead of the usual filtering of navigation information, a filtering of errors in the navigation information is used. Experimental verification shows that because the error is a small amount, it is more accurate when it is linearized. The fusion positioning algorithm can effectively suppress the problems caused by multipath effects and non-line-of-sight (NLOS) in UWB positioning and errors caused by IMU solving in a short time, thus improving the stability and positioning accuracy of the positioning system. The UWB positioning error is stable over a long period, while the inertial navigation positioning error is accurate over a short period to complement each other, overcoming the inability of a single navigation system to achieve precise positioning and navigation. It allows mobile equipment such as coal miners to be better positioned and oriented in GPS denial scenarios such as underground coal mines.

## Figures and Tables

**Figure 1 micromachines-14-01481-f001:**
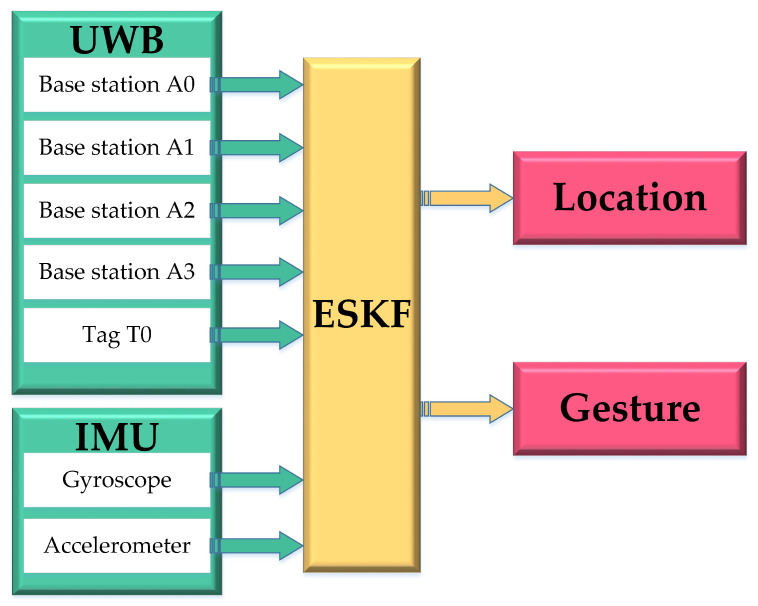
Data-tight coupling method for IMU-based UWB positioning system.

**Figure 2 micromachines-14-01481-f002:**
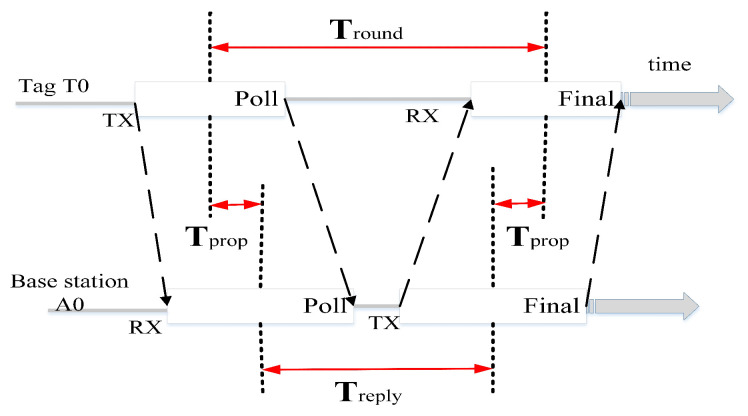
Single-sided two-way ranging.

**Figure 3 micromachines-14-01481-f003:**
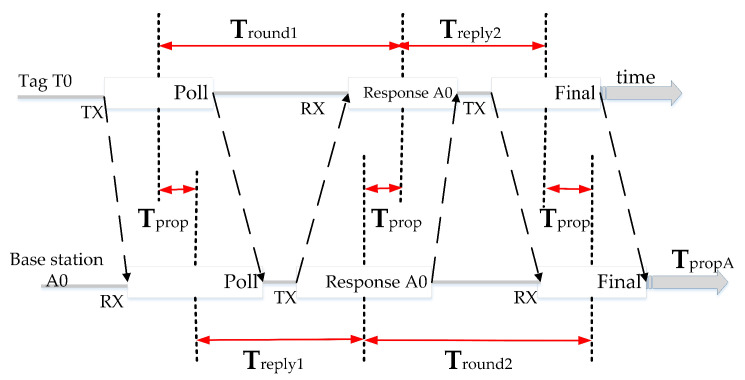
Double-sided two-way ranging.

**Figure 4 micromachines-14-01481-f004:**
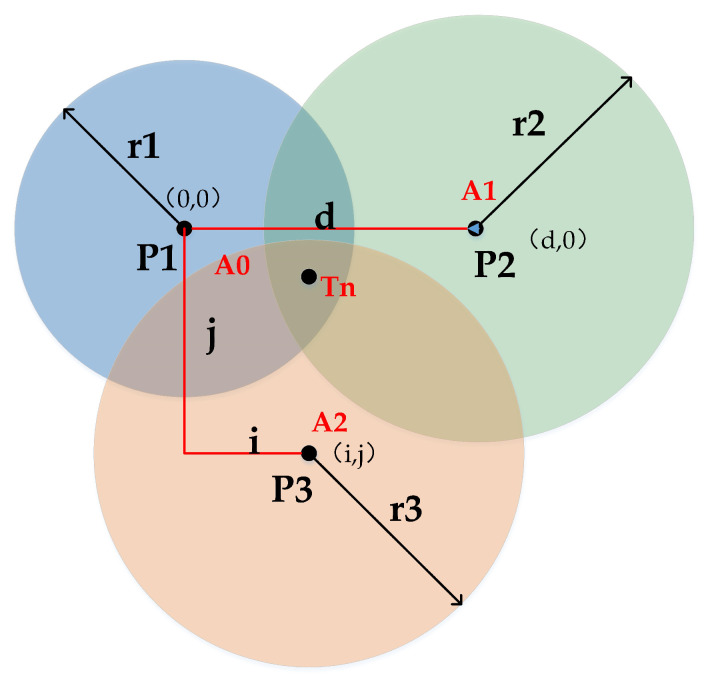
Schematic diagram of the classic trilateral ranging method for UWB positioning.

**Figure 5 micromachines-14-01481-f005:**
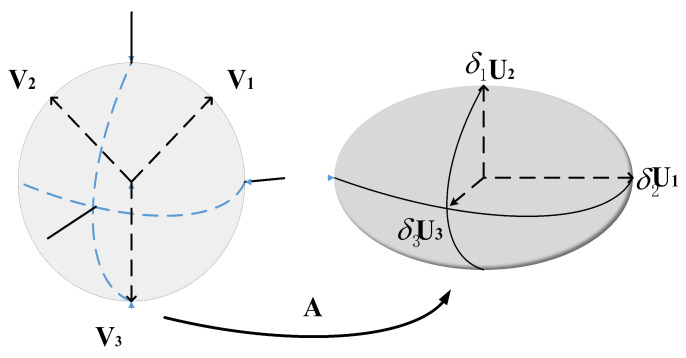
Error accumulation diagram of UWB positioning system.

**Figure 6 micromachines-14-01481-f006:**
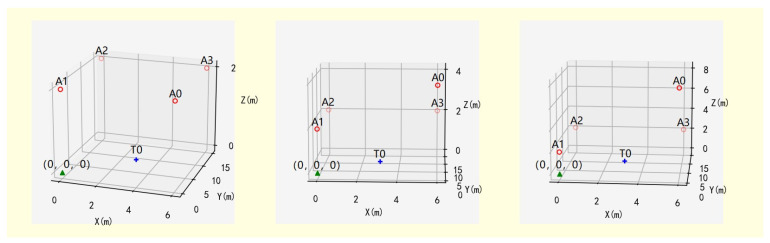
Three base station layouts for IMU-assisted UWB positioning systems.

**Figure 7 micromachines-14-01481-f007:**
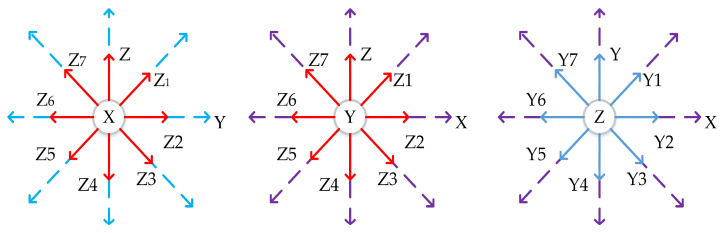
IMU posture placement chart.

**Figure 8 micromachines-14-01481-f008:**
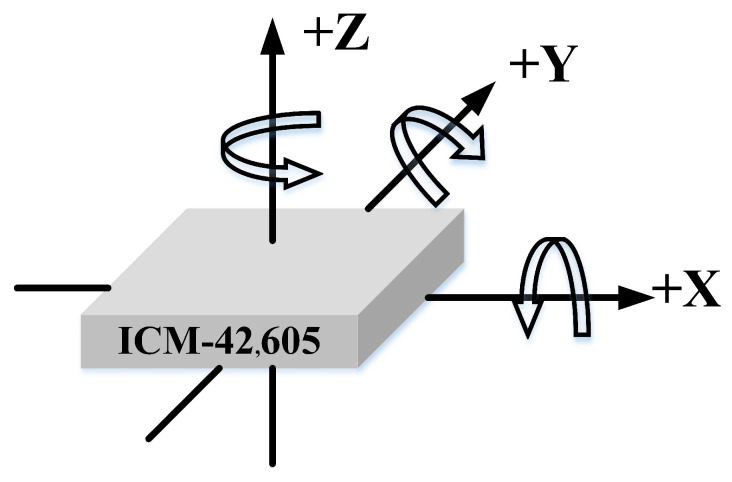
MEMS IMU rotation direction diagram.

**Figure 9 micromachines-14-01481-f009:**
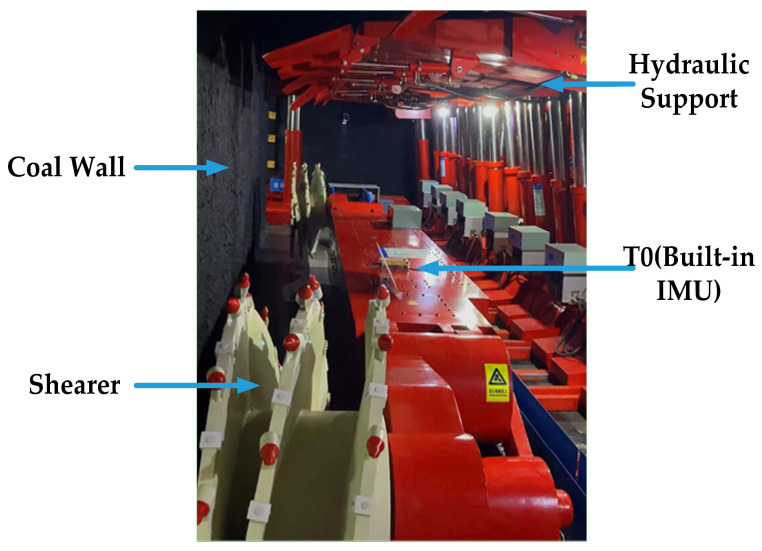
The experimental platform of header mining simulation working face.

**Figure 10 micromachines-14-01481-f010:**
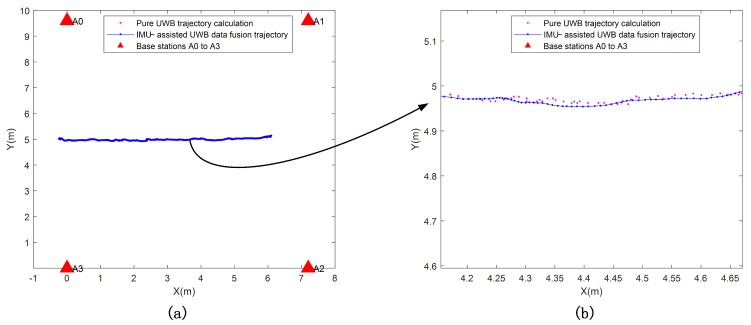
(**a**,**b**) Comparison of the shearer motion trajectory data before and after fusion.

**Figure 11 micromachines-14-01481-f011:**
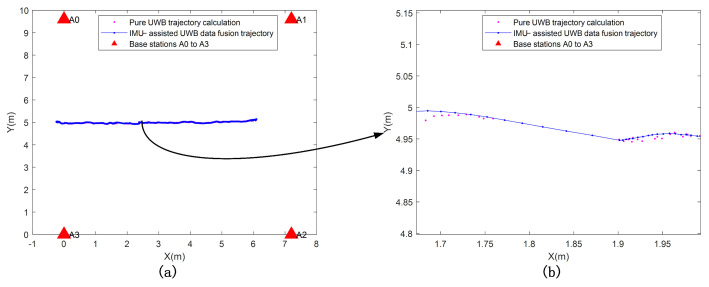
(**a**,**b**) UWB positioning data loss IMU continues to work.

**Figure 12 micromachines-14-01481-f012:**
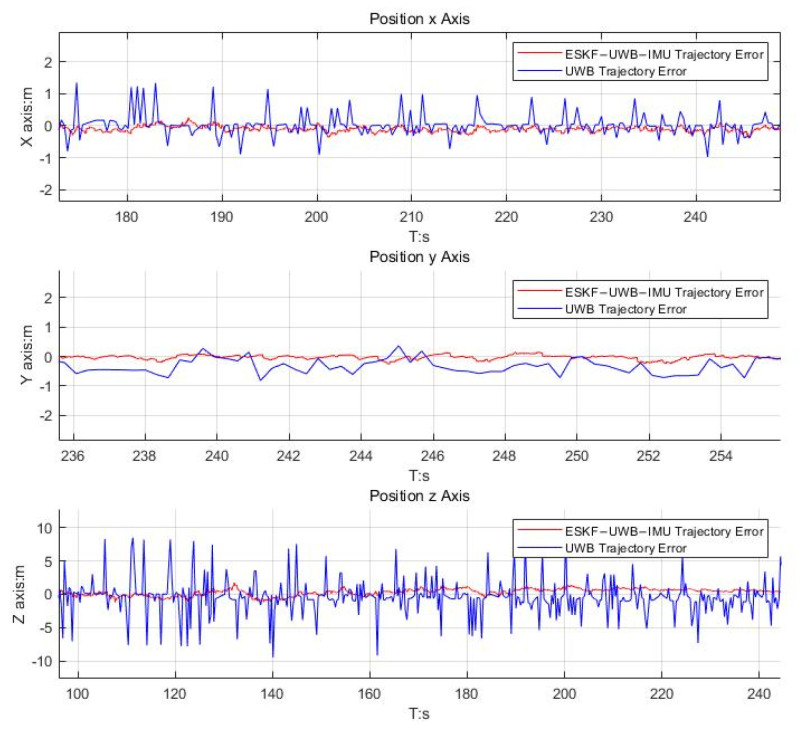
Comparison plot of the position errors between ESKF−UWB−IMU and UWB.

**Figure 13 micromachines-14-01481-f013:**
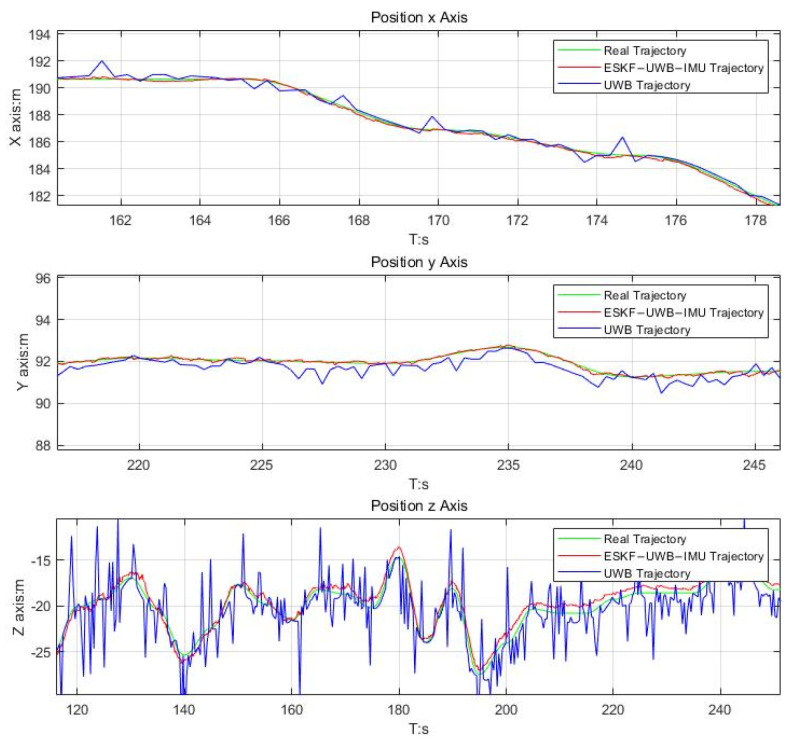
Comparison plot of the position between ESKF−UWB−IMU and UWB.

**Figure 14 micromachines-14-01481-f014:**
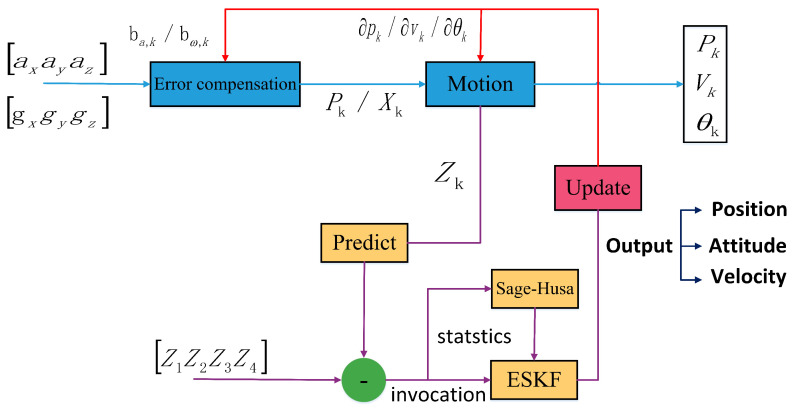
Block diagram of IMU-assisted UWB positioning system flow.

**Table 1 micromachines-14-01481-t001:** Three base station layouts for IMU-assisted UWB positioning systems.

Ellipsoid Axis Length	X-Axis Radius/m	Y-Axis Radius/m	Z-Axis Radius/m
Layout 1	0.423	0.060	0.032
Layout 2	0.162	0.061	0.031
Layout 3	0.088	0.059	0.030

## Data Availability

Data sets cannot be made public for copyright reasons.

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
