# Peer review of "Research on IMU-Assisted UWB-Based Positioning Algorithm in Underground Coal Mines"

_micromachines, 2023, doi:10.3390/mi14071481_

Round 1
Reviewer 1 Report
The authors describe a method of positioning for coal mining equipment in GPS-denied mining environments. A combination of data from a inertial measurement unit (IMU) and ultra-wideband (UWB) ranging technology is used to provide robust and accurate positioning. The authors present two schemes in this context:
Firstly, a method of UWB base station placement optimization, using the singular value decomposition (SVD) of an analytical least squares solution of range measurements and base station coordinates.
Secondly, an error-state Kalman filter to fuse IMU and UWB data. The authors employ additional non line-of-sight (NLOS) detection using a Chi-Squared test as well as an analysis of the UWB signal energy characteristics.
The authors present a plot of data from UWB and from IMU and UWB fusion to show the merit of their positioning scheme.
The authors provide a sound motivation for their approach and the employed methods to solve the presented issue seem appropriate. The overall content of the article is substantial.
However, the description of the methods is severely lacking in clarity and is in some cases contradictory. This reduces any attempt to reproduce or fully review the authors’ methods to guesswork. The following points give details regarding the issues of this article:
1. Besides general spelling, the wording should be reviewed. E.g.: non-visual processing -> Non line of sight processing; solution induced errors (?); measurement-distance […] is negligible -> error of distance-measurement […] is negligible; three-sided ranging -> trilateral ranging; this thesis -> this article/work; “there are various options for the amount of state to be estimated” -> “there are various options for the estimation of the state”;
2. The initial paragraph in section 2.2 is an explanatory sentence from the article template.
3. “[…] uses a trilateral ranging and positioning algorithm known to measure the coordinate values of all base stations in the same reference coordinate system.” Makes no sense in the presented context. The measurement determines the position of a mobile transmitter, not the base stations.
4. It is unclear what is meant “interest” in “The new interest in the ESKF is an […] white noise sequence […].”; In the same section it is unclear what is meant by “wild value” and “wild value processing”
5. The authors describe a UWB TDoA scheme with a need for base station clock synchronization in Section 2. However, in Section 2.1 a bilateral ranging scheme measuring round trip times (RTT) is described. It is thus unclear what kind of scheme was actually used and why clock synchronization is needed. Due to the limited description and the missing derivation of formula 1, it is hard to comprehend the actual ranging method employed by the authors. A more detailed description is needed here.
6. In section 2.2 “c denotes the constant associated with the coordinates of the base station.” It is unclear what this constant is representing.
7. “pseudo distance” is used for the first time in section 2.3 without explanation of the context. If the ranging scheme is based on RTT, the distance measurements should be proper measurements opposed to pseudo distances obtained using TDoA with unknown clock offset between mobile tag and base stations.
8. In section 2.3 an error analysis scheme to judge base station placement using the SVD of the least squares solution from distance estimates and base station coordinates is presented. The authors should give details as to how this method compares to the commonly used analysis of dilution of precision (DOP) known from GPS and more recently UWB applications. Here, the authors should also introduce the variable for the singular value (here: delta) before or right after using it in formula 8. Figure 5 shows the unit vectors “V” transformed by A into U. However, is the transformation not done by the decomposed matrix M (which is derived from A)? Additionally, the vectors U are shown as non-unit vectors. However, they should be shown as unit vectors that are scaled by the singular values delta.
9. In Section 2.3 the error analysis scheme is evaluated. Table 1 shows error the derived error radius along each axis for base station layouts of differing height (z-axis). How can these values be interpreted? Where are these error values valid? At Ground level (Z=0)? After all, the positioning error of a transmitter that is at the same height as the base stations should not change with the height of the base stations (I.e. Z_tag = Z_bs). Can this method really be used to judge the quality of positioning based on the base station placement? Additionally, Figure 6 lacks a legend (circle and triangle) and an explanation for the differing position of the blue triangle.
10. In 2.5 it is unclear what parameter x, y, h and R are representing and how they are transformed into C and b.
11. In section 3 the ESKF is claimed as the “most widely used variant of the Kalman filter” without reference.
12. Formula 17 is missing the transpose sign
13. It is unclear how the matrix of formula 21 is formed and what omega and L are representing
14. It is unclear how deltaP, i.e. the displacement error is determined from the UWB measurement in formula 23.
15. G_t = [I_{3x3} I_{3x12}] should probably be G_t = [I_{3x3} 0_{3x12}]
16. v_k used but not introduced in section 3.4
17. The non line of sight processing scheme based on a Chi-squared test is hard to comprehend due to wording issues and missing introduction of variables: What are r_i and z(k) representing? What is meant by the “new interest in the ESKF”? What is a “wild value”? hat is meant by “sigma gate”?
18. The adaptive noise covariance estimation in section 3.5.1 is missing a description of variables delta z_k and delta x^minus_k. How are these derived?
19. In section 4.1 the “working face space length is 12.94×2.2mm”. Should it be meters, instead of millimeters?
20. Figure 11 mentions pseudopitch without any explanation of the term.
21. The experiment description in section 4 is missing details about the count of test runs. An analysis of the singular test run that shows movement along 7 meters in figures 11 and 14 hardly allows for convincing findings.
22. The experiments are missing a ground truth and a quantification of the positioning error to judge and compare the benefit of their method.
23. In section 4.2 the authors claim that the results show a benefit of their method especially is spaces with low vertical space. However, the experiments lack data for comparison with other methods. As such, this claim can not be backed up by the presented data.
24. The authors mention that the presented method can meet the positioning accuracy requirements for miners. What are these requirements?
25. The authors claim that the ESKF approach is shown to be superior to fusion schemes. However, the article lacks a comparison of these schemes to back up this claim. A experiment with fusion using e.g. an extended Kalman filter is needed.
26. The authors claim that multipath and NLOS is successfully mitigated. However the article lacks the experimental data to back up this claim. A comparison without the presented mitigation schemes is needed.
27. The authors state that in the Data Availability Statement that “All relevant data can be obtained in this article.” However, the experiments can not be reproduced without access to the original UWB and IMU data.
Reviewer 2 Report
1. In section 2.4 the authors suggest that the geometry of the base stations has an effect on the dilution of precision (DOP) of the UWB localization. As DOP is well known to trilateration methods there was no real suggestion from the authors to address this issue with respect to the coal mine application.
2. Please refer to annotation in paper for additional comments.

Please check meaning of technical term for correct interpretations.
Author Response
请参阅附件。

Reviewer 3 Report
Dear Authors:
1- The article looks good and well written.
2- The title reflects the work done.
3- The abstract is good.
4- However, a comparison table should be added before the conclusion part with related work, to show the pros and cons.
5- References are relevant and up to date.
